# Evidence to Underpin Vitamin A Requirements and Upper Limits in Children Aged 0 to 48 Months: A Scoping Review

**DOI:** 10.3390/nu14030407

**Published:** 2022-01-18

**Authors:** Lee Hooper, Chizoba Esio-Bassey, Julii Brainard, Judith Fynn, Amy Jennings, Natalia Jones, Bhavesh V. Tailor, Asmaa Abdelhamid, Calvin Coe, Latife Esgunoglu, Ciara Fallon, Ernestina Gyamfi, Claire Hill, Stephanie Howard Wilsher, Nithin Narayanan, Titilopemi Oladosu, Ellice Parkinson, Emma Prentice, Meysoon Qurashi, Luke Read, Harriet Getley, Fujian Song, Ailsa A. Welch, Peter Aggett, Georg Lietz

**Affiliations:** 1Norwich Medical School, University of East Anglia, Norwich Research Park, Norwich NR4 7TJ, UK; c.nwabichie@uea.ac.uk (C.E.-B.); j.brainard@uea.ac.uk (J.B.); j.fynn@uea.ac.uk (J.F.); amy.jennings@uea.ac.uk (A.J.); b.tailor@uea.ac.uk (B.V.T.); Asmaa.abdelhamid@uea.ac.uk (A.A.); calvin.coe@uea.ac.uk (C.C.); l.esgunoglu@uea.ac.uk (L.E.); C.Fallon@uea.ac.uk (C.F.); tinanana@hotmail.co.uk (E.G.); c.hill2@uea.ac.uk (C.H.); stephanie.howard@uea.ac.uk (S.H.W.); n.narayanan@uea.ac.uk (N.N.); titi.oladosu@doctors.org.uk (T.O.); or emmaprentice557@outlook.com (E.P.); luke.read@uea.ac.uk (L.R.); harrietgetley@hotmail.co.uk (H.G.); fujian.song@uea.ac.uk (F.S.); a.welch@uea.ac.uk (A.A.W.); 2School of Environmental Sciences, University of East Anglia, Norwich Research Park, Norwich NR4 7TJ, UK; n.jones@uea.ac.uk; 3School of Health Sciences, University of East Anglia, Norwich Research Park, Norwich NR4 7TJ, UK; ellice.parkinson@uea.ac.uk; 4Department of Medicine, Luton and Dunstable Hospital NHS Foundation Trust, Lewsey Road, Luton LU4 0DZ, UK; meysoon.qurashi2@nhs.net; 5Lancashire School of Postgraduate Medicine and Health, University of Central Lancashire, Preston PR1 2HE, UK; profpjaggett@aol.com; 6Human Nutrition Research Centre, Newcastle University, Newcastle upon Tyne NE2 4HH, UK; georg.lietz@ncl.ac.uk

**Keywords:** scoping review, vitamin A, infant, child, carotenoids, upper limits, recommended dietary allowances, nutritional requirements, retinol, World Health Organization

## Abstract

Vitamin A deficiency is a major health risk for infants and children in low- and middle-income countries. This scoping review identified, quantified, and mapped research for use in updating nutrient requirements and upper limits for vitamin A in children aged 0 to 48 months, using health-based or modelling-based approaches. Structured searches were run on Medline, EMBASE, and Cochrane Central, from inception to 19 March 2021. Titles and abstracts were assessed independently in duplicate, as were 20% of full texts. Included studies were tabulated by question, methodology and date, with the most relevant data extracted and assessed for risk of bias. We found that the most recent health-based systematic reviews and trials assessed the effects of supplementation, though some addressed the effects of staple food fortification, complementary foods, biofortified maize or cassava, and fortified drinks, on health outcomes. Recent isotopic tracer studies and modelling approaches may help quantify the effects of bio-fortification, fortification, and food-based approaches for increasing vitamin A depots. A systematic review and several trials identified adverse events associated with higher vitamin A intakes, which should be useful for setting upper limits. We have generated and provide a database of relevant research. Full systematic reviews, based on this scoping review, are needed to answer specific questions to set vitamin A requirements and upper limits.

## 1. Introduction

Vitamin A deficiency is a major health problem for many children in low- and middle-income countries. While vitamin A deficiency prevalence has fallen from 39% of children aged 6 to 59 months in low- and middle-income countries in 1991 to 29% in 2013, prevalence remains high in sub-Saharan Africa (48%) and South Asia (44%) [1]. While deaths due to deficiency have been reduced in areas with successful vitamin A programs, 2/3 of countries have no vitamin A deficiency prevalence data from the past decade on which to base nutrient guidelines [2]. 

A recent Cochrane systematic review [3] found that in populations at increased risk of deficiency, oral vitamin A supplementation (using doses of 50,000 to 200,000 IU) in children aged 6 months to 5 years reduced all-cause mortality (RR 0.88, 95% CI 0.83 to 0.93; 1,202,382 participants; high-quality evidence), mortality due to diarrhoea (RR 0.88, 95% CI 0.79 to 0.98; 1,098,538 participants; high-quality evidence), risk of diarrhoea (RR 0.85, 95% CI 0.82 to 0.87; 15 studies; 77,946 participants; low-quality evidence) and risk of measles (RR 0.50, 95% CI 0.37 to 0.67; 6 studies; 19,566 participants; moderate-quality evidence). Another systematic review carried out an individual patient data meta-analysis and found that vitamin A supplementation (doses of 25,000 to 50,000 IU) given within a few days of birth did not affect survival to 6 or 12 months of age [4]. Supplementation was effective in specific settings (trials conducted in southern Asia, in those with moderate or severe vitamin A deficiency, or higher infant mortality rates). However, infant mortality was not reduced with neonatal supplementation in trials conducted in Africa (RR 1.07; 95% CI 1.00 to 1.15) [4], and a further review reiterated that neonatal vitamin A supplementation did not reduce all-cause mortality [5].

Vitamin A is available in two main forms, as provitamin A carotenoids (including beta-carotene, found in fruits and vegetables) and preformed vitamin A (including retinol and retinyl esters, found in animal foods, and used for supplementation programmes). As absorption and conversion of pro-vitamin A carotenoids to vitamin A is variable, consumption of a plant-rich diet may provide insufficient vitamin A [6,7]. Retinol equivalents provide a combined measure of dietary carotenoids and preformed vitamin A, taking account of imperfect carotenoid conversion, though the appropriate conversion factor is debated [6,8]. Status of vitamin A cannot be adequately determined by measuring plasma retinol, since it is homeostatically maintained across a range of intakes. However, when liver vitamin A reserves fall too low, plasma retinol concentrations <0.7 μmoL can be used as an indicator of deficiency, once inflammation has been assessed [9]. Vitamin A stores, either as total body stores, or liver depots can be assessed by biopsy or estimated by the retinol isotope dilution (RID) technique [10].

Nutrient requirements may be calculated using approaches that link intakes with health outcomes (health-based or dose–response approaches) or by calculating and combining data on intake, absorption, conversion, needs for function and growth, depots, and obligatory losses (the modelling or factorial approach). While older nutrient guidelines were based on assessing levels of intake that eliminated signs of deficiency [11,12], modelling approaches have been used in recent decades. For example, US average intakes (AIs) for vitamin A were set for infants according to vitamin A levels in breast milk, and in older children using a modelling approach that included an allowance for adequate liver stores [6], while Nordic guidelines derived children’s vitamin A requirements by extrapolating from adult requirements [11]. The retinol isotope-dilution (RID) technique has also been used to assess retinol intakes needed to maintain status [13]. As vitamin A is stored in the liver, there is a potential for toxicity, so safe upper intake levels (UL) need to be considered as well as minimum requirements. Toxicity has been defined as “A change in morphology, physiology, growth, development, reproduction, or lifespan of a cell, organism, system, or (sub) population that results in an impairment of functional capacity, and impairment of the capacity to compensate for additional stress, or an increase in susceptibility to other influences” [14]. 

Given the importance of vitamin A, its changing deficiency patterns and that a significant amount of new evidence/data has been generated since the FAO/WHO nutrient intake values were last updated [9], a scoping review was undertaken. We scoped the literature to inform the updating of the Food and Agriculture Organization of the United Nations (FAO) and World Health Organization (WHO) nutrient requirements and upper limits for vitamin A for children aged 0 to 48 months [9]. We aimed to identify, quantify, and map the types and sources of evidence available, and thus identify gaps in existing research.

## 2. Materials and Methods

Methods for these scoping reviews were based on Cochrane, using Covidence and Microsoft Excel software [15,16], reported according to PRISMA-ScR guidelines [17]. The review protocol was submitted to the WHO as part of our funding bid (available from the authors on request). Two main changes have occurred since submission:
Revision to search systematically for and include children aged 0 to 48 months, but also include any relevant studies identified in infants and children aged up to 10 years (mean age ≤120 months), so that relevant studies, that may be scaled for younger children, could be included (WHO originally requested inclusion of studies on children aged 0 to 36 months).The WHO requested that we search from the inception of each database (rather than from 2010 onwards, as suggested in the protocol). 


This broadening of our remit was not accompanied by an increase in the resources provided, which meant that we could not collect the full texts of all potentially relevant studies (as earlier research is less accessible).

The questions set out within the protocol are shown in Box 1. These questions all relate to children aged 0 to 48 months in any geographical location. Details of specific nutrient biomarkers, bioavailability, excretion, body stores or depots, etc. were taken from recent guidance [8]. We considered the types of studies that would help to answer both the health-based and modelling-based questions in setting the inclusion criteria. The inclusion criteria are set out in full in Appendix A.

Box 1Review questions informing our inclusion criteria.Health-based questions for vitamin A:What is the relationship between exclusive or mixed breastfeeding duration and vitamin A status?What is the relationship between duration of formula use and vitamin A status?What is the relationship between vitamin A intake (from formula, foods and sup-plements) and any health outcome?What is the relationship between vitamin A intake (from formula, foods and sup-plements) and vitamin A status (such as serum retinol and liver stores)?What is the relationship between vitamin A status and any health outcome (such as night blindness, xerophthalmia, diarrhoea, infection mortality, all-cause mortality, infection rate, measures of growth)?
Modelling based questions for vitamin A:What are the obligatory losses of vitamin A in exclusively breast-fed infants, infants on mixed feeding (breast and formula), infants on breast milk and weaning foods, infants on formula and weaning foods, infants on follow-on milk and weaning foods, and fully weaned children?What are vitamin A requirements for growth and storage in infants and children?How large are vitamin A stores and total body vitamin A pools at different ages?How well are carotene and pre-formed vitamin A from breast milk and infant for-mula, from specific weaning and other foods, supplements, fortified foods and bio-fortified foods, absorbed?What evidence do we have on levels of conversion of carotenoids to functional vita-min A in children aged 6 to 48 months?How is carotene conversion linked to vitamin A status?


### 2.1. Searches

We developed complex electronic searches using text and indexing terms (these are called MeSH terms in Medline), truncation and controlled language. Searches were run on Medline (Ovid), EMBASE (Ovid), and Cochrane Central, from inception to 19 March 2021, based on the format:

[vitamin A intake or status] and [infants or young children] and [human]

As we were awarded contracts for three scoping reviews (for magnesium, iron and vitamin A) and there was considerable overlap between the results of the searches for each nutrient, the search strategies were adapted to include all three nutrients (full texts of the searches are presented in Appendix A). Search strategies were not limited by language, methodology or health outcomes to ensure complete results, including novel outcomes. We used previous guidelines developing dietary reference values (DRVs) and upper limits (ULs) [6,8,9,11,12,18,19,20,21,22], to help identify key studies, evidence assessments and methods of analysis. Our subject expert (GL) was asked to check the database of studies for gaps and identify any particularly useful studies for guideline production. 

### 2.2. Assessment of Inclusion

Titles and abstracts from electronic searches were uploaded into Covidence software (Veritas Health Innovation, Melbourne, Australia, available at www.covidence.org (accessed on 31 December 2021). A training set of 222 titles and abstracts was created, assessed and then discussed by the entire review team to ensure a consistent approach. Inclusion assessments were carried out independently in duplicate, disagreements were appraised at weekly meetings and by a third reviewer (LH) where needed. The review team, including topic experts, met weekly (virtually, with detailed circulated minutes) to discuss inclusion decisions and clarify inclusion criteria. Full texts of potentially relevant studies were located and added to Covidence. Some full texts were unavailable, so where inclusion was not clear from the abstract, these studies were retained in our database for future assessment. 

Assessment of inclusion of full texts was completed independently in duplicate for 20% of studies, with remaining studies assessed singly. This was a change to our original protocol, made necessary by the large number of studies derived from the search strategy. Our expert panel determined this to be efficient and acceptable due to the high inclusion rate and low disagreement rates by the reviewers. 

### 2.3. Data Extraction and Tabulation

Potentially relevant studies were included in the scoping reviews, tagged in Covidence by nutrient, question and study design. Studies were tabulated with bibliographic details, title, abstract (where available), publication year, with additional data extraction and a risk of bias assessment for some studies (see below). 

We created separate spreadsheet tables (in Microsoft Excel) for each nutrient (vitamin A, magnesium, iron). Within each we created separate sheets for each relationship with these titles: Intake Outcome; Intake Status; Status Outcome; Factorial Relationships; and Adverse effects/Toxicity/Overload. Each table was split by study design: Systematic reviews; Isotopic studies; randomised controlled trials (RCTs) and other trials; Cohort and Case control studies; Cross sectional studies; and Non-systematic reviews (collected to help gather further primary references for any future systematic reviews) and ordered by publication year. Recent studies were defined as those published since January 2013 (2 years before the European Food Safety Authority (EFSA) guidance [8]), and highlighted in the Excel sheets for emphasis.

As a critical question for the commissioning of the WHO guidelines group is whether to move to the intake-health model from the factorial approach to set DRVs; we focused on assessing intake-health data. Additional data extraction was carried out for some studies:(a)Intake-status-outcome studies (Appendix A): We undertook limited data extraction to clarify available outcomes (e.g., mortality, growth, infections), adverse effects and sample sizes for recent systematic reviews and trials.(b)For outcomes assessed in ≥6 trials, or trials including at least 1000 children, we carried out additional data extraction on relevant trials. This second layer of data extraction included:i.Interventions (e.g., dose, frequency, duration & type of vitamin A plus whether further nutrients were included in the intervention)ii.Details on participant age, country, and baseline health statusiii.How the outcome was measurediv.Allocation method


Adverse effect, toxicity and overload studies: where systematic reviews and trials assessing intake-health (Dataset, can see Appendix A) reported adverse effects or toxicity in some way, these studies were copied into the adverse effects sheet (Dataset, can see Appendix A).

### 2.4. Risk of Bias Assessment

As this is a scoping review, we did not carry out detailed risk of bias assessment for most of the included studies (this would be appropriate in a focused systematic review). However, as they are crucial, we did carry out rapid risk of bias assessment using Amstar [23] for relevant systematic reviews of intake-status-outcome studies (Dataset, can see Appendix A). Allocation method was also noted for particularly relevant RCTs (as above).

## 3. Results

### 3.1. Search Results

Electronic searches retrieved 48,747 titles and abstracts of potentially relevant studies on magnesium, iron and vitamin A, reduced to 35,347 on de-duplication and merging of papers into studies (Figure 1). Of these, 30,146 were excluded. The remaining 5201 papers underwent full-text assessment, but full texts could not be obtained for 775, of which 278 were potentially relevant studies of vitamin A (see Dataset Excel sheet 1 Awaiting Assessment, which is a list of studies that could be obtained in full text and re-assessed for inclusion for any future full systematic review). A total of 1251 were excluded (with reasons, see Figure 1), and 3175 included for one or more of the iron, vitamin A or magnesium scoping reviews; 899 contributed information on the topic of vitamin A and are represented in the Excel database (Figure 1). Some studies appear on several sheets. 

### 3.2. Data Relevant to Setting Dietary Reference Values (DRVs)

#### 3.2.1. Intake Outcome Relationships

Studies assessing the relationship between vitamin A intake and health, growth or development outcomes are key to the health-based method of setting dietary reference values. These studies are found in Excel sheet 2A Intake Outcome. We identified 18 recent systematic reviews (published since early 2013, shown with data extraction and risk of bias assessment), 26 older systematic reviews, 43 recent RCTs (with data extraction), and 134 earlier trials. Additionally, nine recent and 26 older cohort and case–control studies, 14 recent and 21 older cross-sectional studies and seven recent and 23 older non-systematic reviews are noted. 

Studies assessing the relationship between vitamin A intake and a marker of vitamin A status (Excel sheet 2B Intake Status) and between status and health, growth, or development outcomes (Excel sheet 2C Status Outcome) may in combination support the data directly assessing intake and outcomes. For Intake Status studies, we included three systematic reviews published since early 2013 [24,25,26], which were data-extracted and assessed for risk of bias (plus two older systematic reviews), 31 recent trials, of which 12 appeared particularly recent and relevant [27,28,29,30,31,32,33,34,35,36,37,38] (and 80 older trials). Alongside these we noted 12 recent cohort or case–control studies, 15 earlier studies, nine recent cross-sectional studies, 25 older studies, and nine non-systematic reviews. For studies assessing the relationship between vitamin A status and health, growth or development outcomes we identified no systematic reviews, two recent RCTs (with additional data extraction), 39 older RCTs, 17 recent observational studies of which eight appeared particularly relevant [39,40,41,42,43,44,45,46], 42 earlier cohort or case–control studies, 19 recent cross-sectional studies, 56 earlier cross-sectional studies and eight potentially relevant non-systematic reviews.

#### 3.2.2. Evidence Addressing Health-Based Questions for Vitamin A

The evidence addressing health-based questions, including the number of each type of study and references of key papers, is summarised in Table 1.

a.What is the relationship between exclusive or mixed breastfeeding duration and vitamin A status in children?

The most relevant studies measuring vitamin A intake appear on Excel sheet 5 Potentially Useful Reviews. Five relevant systematic reviews, including one undertaken to inform the US 2020 Dietary Guidelines Advisory Committee (search September 2019) [47], one carried out by the USDA Nutrition Evidence Systematic Review Team and Complementary Feeding Technical Expert Collaborative (search March 2016) [48], a Cochrane review (not updated since June 2011) [51], and two further reviews [49,50]. 

b.What is the relationship between duration of formula use and vitamin A status in children?

Three systematic reviews mentioned in the previous section also addressed this question [47,48,50]. 

c.What is the relationship between vitamin A intake (from formula, foods and supplements) in infants and children and any health outcome?

This evidence is found in Excel sheet 2A Intake Outcome. High quality individual patient data meta-analysis and Cochrane systematic reviews assessed the relationship between vitamin A intake and health outcomes. Three assessed the relationship between vitamin A intake (by supplementation) and mortality in the first few days of life (11 trials including 163,000 neonates) [4], in infants aged one to 6 months (12 trials, 24,000 infants) [52] and children aged 6 months to 5 years (47 trials, 1.2 million children) [3]. These same two reviews also assessed effects on cause-specific mortality, morbidity, vision, and side effects [3,52]. Only one of the 18 recent systematic reviews assessed effects of vitamin A sources other than high-dose preformed vitamin A supplements, assessing fortification of staple foods [25] (as did one of the 26 older systematic reviews, assessing agricultural interventions [53]). Recent trials report effects of increasing vitamin A intake in infants and children on mortality and a variety of types of morbidity such as immune response [28,29,54,55,56], atopy [57,58], respiratory infection [59], cognition [60,61], eye health [33,38,62] and growth [38,63]. Most of the 43 recent trials assessed effects of supplementation (though two assessed effects of complementary foods, one alongside home fortification [64,65], two biofortified maize [33,62], one biofortified cassava [66], one carotenoid enriched juice [67], and one fortified milk [68]). 

d.What is the relationship between vitamin A intake (from formula, foods and supplements) and vitamin A status (such as serum retinol and liver stores)?

The most relevant studies appear on the Excel sheet 2B Intake Status. There are three relevant systematic reviews [24,25,26], plus a set of trials assessing effects of supplementation on serum retinol and beta-carotene. Many isotopic studies (shown in Table 2) also assessed intake status relationships. The majority of the 31 recent trials assessing effects of vitamin A intake on vitamin A status measures (Excel sheet 2B intake status), focused on supplementation. Fifteen trials assessed effects of biofortified cassava [27,34,66], sweet potato [69] or maize [31,33], complimentary foods [70], peanut butter and kale [71], high-carotenoid juice [67], different infant formulae [35,37], fortified rice [36], cow peas and amaranth [72], and home fortification with multiple micronutrient powder [65,73].

e.What is the relationship between vitamin A status in infants and children and any health outcome (such as night blindness, xerophthalmia, diarrhoea, infection mortality, all-cause mortality, infection rate, measures of growth)?

As expected from the nature of the question, most of the studies available to address the relationship between vitamin A status and health outcomes were observational, assessing relationships between markers of vitamin A status and autism spectrum disorders [39], acute or recurrent respiratory infection [40,43,44,46], asthma [42], malaria [41], infectious diseases generally [45] and mortality [41] (Excel sheet 2C Status Outcome)**.**


#### 3.2.3. Factorial Relationships

We originally separated out studies on vitamin A absorption, stores, losses and excretion, needs and metabolism, and balance. However, we discuss them together as there is a great deal of overlap (Excel sheet 3 Factorial). The most relevant systematic review (“Metabolism of Neonatal Vitamin A Supplementation” [24]) focused on the first 28 days of life. The authors of that review searched systematically between August 2013 and 5 Jan 2020, with a supplemental Medline search extending to January 2020. Included studies were of neonatal humans and animals, given single or periodic oral vitamin A (less than daily). Outcomes were absorption (five human studies assessed short term serum response, four unabsorbed vitamin A which is most likely not a reliable measurement of vitamin A absorption due to degradation of vitamin A via the microbiome), transport (no human studies), storage (one human study), metabolism and detoxification (no human studies) and organ maturation (one human study). All their included human studies were published before 1995, and almost all before 1960. 

Our review identified 15 newer isotopic studies (published alongside their trials registry entries and conference abstracts, since early 2013), which between them addressed absorption, metabolism, balance, body depots and excretion (see Table 2 [32,36,68,74,75,76,77,78,79,80,81,82,83,84,85,86,87,89,90], alongside nine older isotopic studies [88,91,92,93,94,95,96,97,98]). The potentially most relevant recent trials explored the effects of multiple nutrient supplementation [99,100,101,102,103], vitamin A supplementation on iron metabolism [104] or supplementation in conjunction with other treatments [105]. Older trials assessed the effects of vegetables and green leaves [96,106], milk formula [107], fortified seasoning powder [108], micronutrient supplement [109] and high-dose supplementation [110,111,112,113,114,115,116,117,118]. The observational studies and non-systematic reviews are noted on Excel sheet 3. 

#### 3.2.4. Methodologies Used in Previous DRV Development

The methods used to develop DRVs in previous guidelines are included in Appendix A. The table of references used within previous guidance, and their context, appears in Appendix A. Guidelines tend to cite previous guidelines (Appendix A). Note that the text of Appendix A are largely “cut and paste”—for information only. 

### 3.3. Data Relevant to Setting Upper Limits (ULs)

#### 3.3.1. Studies on Vitamin A Adverse Effects, Toxicity, and Overload

Studies on toxicity and adverse effects of vitamin A in healthy infants and children are shown in Excel sheet 4, Adverse Effects. Some studies were identified as primarily assessing toxicity (for example, acute accidental poisoning), but most were included elsewhere in the review, for example, addressing the relationship between intake and health outcomes, or intake and status, where adverse or negative effects of high vitamin A intakes were assessed. 

Six recent systematic reviews reported adverse effects of vitamin A supplementation or overload [3,5,24,52,119,120], and a useful older systematic review collated case reports on toxicity due to retinol or retinyl esters in foods or supplements [121]. Two recent isotopic studies provided data on hypervitaminosis A [90,122], and 13 recent trials reported adverse effects of vitamin A supplementation [37,38,57,65,123,124,125,126,127,128,129,130]. Additionally, we note 19 recent and 23 older observational (non-isotopic) studies (six of which reported on hypercarotenaemia), plus three recent and three older non-systematic reviews. A subset of studies that may be particularly useful for assessing upper limits has been highlighted by GL (shown at the bottom of Excel sheet 4 Adverse Effects) [121,131,132,133].

#### 3.3.2. Methodologies Used in Previous Vitamin A UL Development

Methods of UL development from previous guidelines are cut and pasted into Appendix A, and references used within previous guidelines to underpin UL derivation appear in Appendix A. 

## 4. Discussion

This scoping review identifies and details the most relevant research for use in updating nutrient requirements and upper limits for vitamin A for children aged 0 to 48 months. A body of new research (published since early 2103, two years before publication of the most recent EFSA opinion [8]) has been published and is available for use in setting guidance, whether a health-based or modelling-based approach is chosen. 

Although this is a scoping review and not a systematic review, we have used systematic methods to identify, quantify, and map research for use in updating nutrient requirements and upper limits for vitamin A in children. However, we acknowledge limitations, including the lack of access to older full text papers, which was due to a lack of resource. We also present only limited information on the risk of bias, methods, and outcomes of each included study. 

### 4.1. Health-Based Approach to Nutrient Requirements

Data mapping suggests that there may be sufficient data to set DRVs using intake outcome and intake status trials, even omitting trials of supplementation. Research assessing effects of vitamin A intakes from breastfeeding, formula feeds, complementary and other foods from a range of cultural settings are potentially most useful, and trials assessing the effects of supplementation on immune response, serum retinol, and beta-carotene could support mortality data. The effects of vitamin A supplementation on mortality have been recently systematically reviewed, with searches run to early 2016. Undertaking a comprehensive systematic review assessing the quantitative relationship between vitamin A intake (in a variety of forms, including usual foods, formula, fortified and biofortified foods, added fortification vitamin A, but not vitamin A supplements) on health, development, growth, adverse events and key (defined) measures of vitamin A status in infants and children (with assessment of basal vitamin A intake and status) would appear useful to underpin health-based guidance. Ideally, primary studies will assess vitamin A intakes from provitamin A carotenoids and preformed vitamin A in breastmilk, formula, complementary foods, supplements and fortified foods when assessing the effects or associations with health outcomes. Further primary studies assessing the effects of quantified vitamin A intake from dietary, fortification and supplementary sources in infants and children on health, development, growth and adverse events would be useful.

### 4.2. Modelling-Based Approach to Nutrient Requirements

Our scoping review identified recent isotope tracer data which are likely to be a good approach for quantifying effects of bio-fortification, fortification and food-based vitamin A on vitamin A total body and liver stores, losses, needs and balance. The identified recent studies should enable the assessment of bioefficacy (the combination of absorption and bioconversion) of provitamin A carotenoids and provitamin A conversion factors under field conditions [8]. Mathematical modelling using “super-person” designs with adequate datasets will allow calculation of the “fractional catabolic rate”, which gives a good indication of daily vitamin A losses, hence vitamin A excretion. Such study results could be incorporated into the final analysis of DRVs for infants and children. 

Vitamin A absorption has traditionally been assessed by measuring levels of excreted vitamin A in faeces and urine, but logistical problems in field and laboratory, and bacterial degradation of retinoids in the microbiome, likely reduce the accuracy of this approach. Measuring serum retinyl ester concentrations after a defined oral dose combined with mathematical compartmental analysis offers a potentially more accurate assessment of vitamin A absorption. Similarly, carotenoid absorption can be estimated from serum concentrations in the chylomicron fraction 6–8 h postprandially after a defined oral dose. Assessing vitamin A absorption from foods in infants and children is ethically and logistically challenging as it requires a series of blood samples taken over 6–8 h. This is an even greater issue in at risk populations. To overcome this issue a super-child design can be used to obtain accurate absorption data across a group of children. For provitamin A carotenoids, variation in absorption and bioconversion both contribute to inter-individual variation. Bioefficacy determination enables assessment of bioequivalence of provitamin A carotenoids from different foods. The Retinol Isotope Dilution (RID) technique, or dual isotopes (labelled preformed retinol combined with labelled provitamin A carotenoids) can accurately assess bioefficacy. A recent approach to assessing vitamin A absorption uses an area under the curve approach, which appears promising in field conditions [134]. 

Vitamin A losses have traditionally been assessed using urinary and faecal losses after defined dose application or during periods of disease. Research using isotope tracers combined with mathematical compartmental analysis also allows determination of the “fractional catabolic rate”, a more accurate assessment of vitamin A losses over time. Any systematic review would need to make these distinctions clear and include future recommendations for assessing vitamin A losses. There is a need for future studies to study the ‘fractional catabolic rate’ during periods of disease. 

The scoping review suggests that data are limited on absorption, conversion, stores, losses, needs and balance of vitamin A from a wide range of normal diets in infants and children. A systematic review of the existing isotopic studies would be useful to clarify details of absorption, metabolism, stores, growth, losses and balance in children of different ages, from different dietary sources in different parts of the world. 

### 4.3. Upper Limits

Understanding the relationship between vitamin A intake and toxicity or side effects is important in order to set appropriate upper limits for vitamin A in infants and children. Systematically reviewing adverse effects reported in relevant efficacy trials as well as trials of negative outcomes would produce a stronger dataset of adverse events. 

## 5. Conclusions

We have produced an extensive dataset of studies that may be relevant in setting vitamin A DRVs and upper limits in infants and young children. We believe this dataset will be useful in helping researchers to focus future research, and underpin systematic reviews on supporting the setting of DRVs and upper limits. Our mapping suggests that there are potentially sufficient studies to set DRVs for infants and young children for vitamin A, using both the health-based and modelling-based approaches. To enable either approach, new or updated systematic reviews of specific sections of the data will be needed. Ideally, both the health-based and modelling-based approaches to setting DRVs would be attempted independently, and the results compared to obtain the most robust DRV estimates. Data for setting upper limits in young children are more limited and may require extrapolation from older children and adult populations. 

## Figures and Tables

**Figure 1 nutrients-14-00407-f001:**
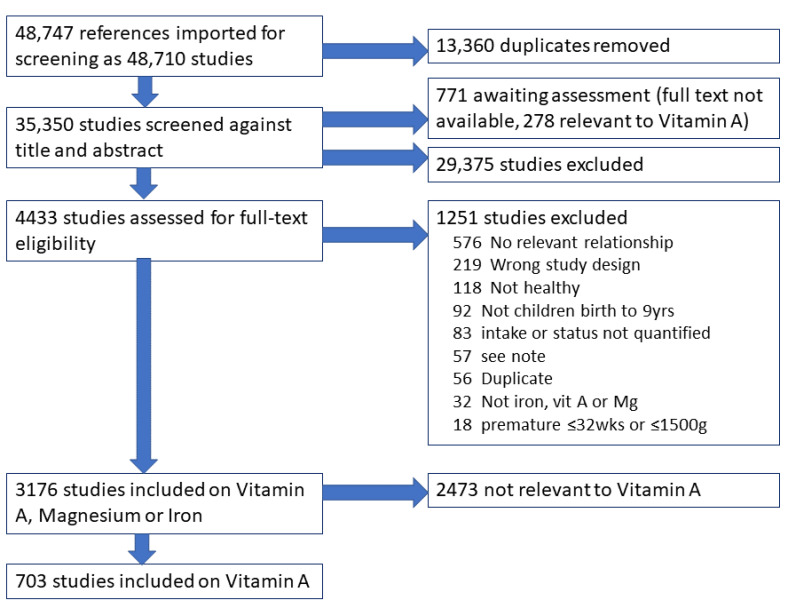
PRISMA flow chart.

**Table 1 nutrients-14-00407-t001:** Mapping of relevant research addressing health-based questions: number of relevant studies of each methodology, plus references to the most relevant studies.

	Systematic Reviews		RCTs & Trials		Cohort & Case-Control Studies		Cross-Sectional Studies	
	2013+	Pre-2013	2013+	Pre-2013	2013+	Pre-2013	2013+	Pre-2013
What is the relationship between exclusive or mixed breastfeeding duration and vitamin A status in children?	4 [47,48,49,50]	1 [51]						
What is the relationship between duration of formula use and vitamin A status in children?	3 [47,48,50]	0						
What is the relationship between vitamin A intake (from formula, foods and supplements) in infants and children and any health outcome? (See Excel sheet 2A Intake Outcome)	18 [3,4,25,52]	26 [53]	43 [28,29,33,38,54,55,56,57,58,59,60,61,62,63,64,65,66,67,68]	134	9	26	14	21
What is the relationship between vitamin A intake (from formula, foods and supplements) and vitamin A status? (See Excel sheet 2B Intake Status)	3 [24,25,26]	2	31 [27,31,33,34,35,36,37,65,66,67,69,70,71,72,73]	80	12	15	9	25
What is the relationship between vitamin A status and any health outcome? (See Excel sheet 2C Status Outcome)	0	0	2	39	17 [39,40,41,42,43,44,45,46]	42	19	56

**Table 2 nutrients-14-00407-t002:** Details of recent isotopic studies.

Study	Country	Vitamin A Source	Children’s Ages	Vitamin A Outcomes Assessed
Ford 2020 [74], NCT03000543 [75], NCT03345147, NCT03030339	Bangladesh, Guatemala, Philippines	Some supplemented, others dietary only	9–65 months	TBS, retinol kinetics
Ford 2020 [76]	Bangladesh, Phillipines, Guatemala, Mexico	Dietary and supplemental intake	Birth to 5 years	TBS, liver concentration
Lopez-Teros 2020 [77]	Mexico	Usual diet & supplementation	3–6 years	Whole-body retinol kinetics, TBS
Lopez-Teros 2017 [78,79]	Mexico	Moringa oleifera leaves	17–35 months	VA equivalence, TBS, retinol kinetics
Lopez-Teros 2017 [80,81]	Mexico	Breast milk	0–2 years	Breast milk intake, VA intake from breast milk
Lopez-Teros 2013 [82], Astiazaran-Garcia 2013 [68]	Mexico	Fortified milk	Pre-school	TBS, SR, liver VA concentration
Mondloch 2015 [83]	Zambia	Biofortified maize	Pre-school	TBS, serum carotenoids, RBP etc
Muzhingi 2017 [71,84]	Zimbabwe	Peanut butter and kale	12–36 months	Conversion factor
NCT03383744 [32]	Cameroon	Supplementation	3–5 years	TBS, SR, RBP
NCT03801161 [85]	Bangladesh	Usual dietary intake	9–18 months	SR, TBS, RBP, beta carotene, CRP, iron status
NCT02363985, NCT03194724, NCT03207308 [86]	Ethiopia, Cameroon, Botswana, Senegal	Dietary diversity, supplementation, biofortification	3–5 years	TBS, SR, Liver stores, infection, dietary intake, anthropometry, morbidity
Palmer 2021 [87], NCT02804490	Zambia	Biofortified or fortified maize to mother	9 months	TBS, breast milk retinol
Pinkaew 2013 [36], NCT01199445 [88]	Thailand	Fortified rice	School age	TBS, SR
Suri 2015 [89], NCT01061307, NCT01814891	Thailand, Zambia	Usual intake and status	Pre-school	SR, total liver reserves
Van Stuijvenberg 2019 [90], NCT02915731	South Africa	Supplementation, fortification, sheep liver intake	Pre-school	Hypervitaminosis A, TBS

SR serum retinol, TBS total body stores, VA vitamin A, RBP retinol binding protein.

## Data Availability

The data presented in this scoping review are available in the associated Excel sheet and Appendix A. The protocol (which formed part of the funding application) is available on request from the authors.

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
