# Peer review of "Evidence to Underpin Vitamin A Requirements and Upper Limits in Children Aged 0 to 48 Months: A Scoping Review"

_nutrients, 2022, doi:10.3390/nu14030407_

Round 1

Reviewer 1 Report

The article is interesting to public health. I suggest reviewing the summary, structuring it in accordance with PRISMA 2020 standards.
The text can benefit from the review with the criteria of PRISMA Scoping reviews.

Tricco, AC, Lillie, E, Zarin, W, O'Brien, KK, Colquhoun, H, Levac, D, Moher, D, Peters, MD, Horsley, T, Weeks, L, Hempel, S et al. PRISMA extension for scoping reviews (PRISMA-ScR): checklist and explanation. Ann Intern Med. 2018,169(7):467-473. doi:10.7326/M18-0850

Author Response

Authors’ reply: Thank you, it is an excellent idea to mention PRISMA-ScR in the methodology rather than just in the associated checklist.  We have reported our scoping review in compliance with PRISMA-ScR (as seen in the PRISMA flow chart (Fig 1) and the associated PRISMA-ScR checklist). We have added a sentence at the beginning of the methods section on general methodology, which includes PRISMA-ScR and its reference.  We have checked the text again and believe that it is structured in accordance with PRISMA-ScR (see the submitted PRISMA-ScR checklist).

Yours sincerely, Lee Hooper (on behalf of all the authors)

Reviewer 2 Report

The manuscript mapped the survey for use in updating nutrient requirements and upper limits for vitamin A in children aged 0 to 48 months. It is exonerated for the state of the art of vitamin A. But, it is confused. ABSTRACT It is necessary to include a conclusion ... what was found important to contribute to the establishment of the nutritional requirement? What is the study mapping application? Do they bring similar contributions on the application? What remains to be done? Keywords: Use words other than the title - see MeSH INTRODUCTION LINE 99-100: include in the previous sentences. It got repetitive here. 125 line: “recently” is 2011? METHODS search why not use mesh descriptors? And their equivalents names? Inclusion of terms like VAD, vitamin A status would apply the results. Inclusion criteria: It seems confusing how the authors arrived at the inclusion criteria. What were the eligibility criteria? Types of studies? To describe. Why include magnesium and iron? Describe how you assessed the risk of bias for each type of study. Include topic. RESULTS Need to include more tables to understand the study DISCUSSION Same results. It ends without the application of this study... The study has no conclusion. REFERENCES Adapt to the magazine's format

Author Response

The manuscript mapped the survey for use in updating nutrient requirements and upper limits for vitamin A in children aged 0 to 48 months. It is exonerated for the state of the art of vitamin A. But, it is confused.

ABSTRACT It is necessary to include a conclusion ... what was found important to contribute to the establishment of the nutritional requirement?

Authors’ reply: The findings of a scoping review are the scope of the research located, and we report here the characteristics of the relevant research located.  It is not appropriate for us (in carrying out a scoping review) to assess the meanings of this research (this would require a focused and analytical systematic review, rather than a scoping review). As you suggest, we have added an appropriate conclusion (page 2, lines 71 to 73 in the clean version).   

What is the study mapping application? Do they bring similar contributions on the application?

Authors’ reply: We used Microsoft Excel to format the dataset (please see the Excel dataset attached to the submission), and state this in the methodology (page 4, lines 146 to 148) though not the abstract as words are limited.  Apologies, we are not sure what you mean when you ask “Do they bring similar contributions on the application?”. 

What remains to be done?

Authors’ reply: What remains to be done is a great question, we have added a sentence to address this at the end of the abstract “Full systematic reviews are needed to answer specific questions to set vitamin A requirements and upper limits” (page 2, lines 71 to 73 in the clean version) and have addressed this more fully in the conclusions.

Keywords: Use words other than the title - see MeSH

Authors’ reply: Keywords relate to indexing (MeSH) terms where appropriate MeSH terms exist. The following are MeSH terms:  vitamin A; infant; child; carotenoids; Recommended Dietary Allowances; nutritional requirements; World Health Organization.  There are no MeSH terms to describe scoping reviews (they are not literature reviews or meta-analyses), retinol or upper limits (which are of course distinct from RDAs) so we used these as indexing terms despite their not being MeSH terms. We believe these terms cover the key concepts of the research.

INTRODUCTION LINE 99-100: include in the previous sentences. It got repetitive here. 125 line: “recently” is 2011?

Authors’ reply: thank you, we have pulled these two sentences together and removed the word “recently”.

METHODS search why not use mesh descriptors? And their equivalents names? Inclusion of terms like VAD, vitamin A status would apply the results.

Authors’ reply: Searching: yes we did include MeSH terms (we call them indexing terms as MeSH terms are specific to Medline, while indexing terms relate to any database including EMBASE).  The full texts of all the searches (including all the indexing terms) are in the supplemental materials, and we have made this clearer in the text.  In the Medline search index terms used included “beta carotene/ or vitamin a/”, and text terms (with truncation) included “(retinol*or retinal* or cryptoxanthin* or caroten* or beta-caroten* or vitamin a or (retinyl adj ester*)).ab,ti.”  These terms were checked and agreed by WHO and reflect the search strategies used in other systematic reviews on vitamin A (1).

Inclusion criteria: It seems confusing how the authors arrived at the inclusion criteria. What were the eligibility criteria? Types of studies? To describe.

Authors’ reply: We have included a fuller description of how the inclusion criteria were developed (lines 164 to 167), and reference Appendix 1, which includes the full set of inclusion criteria here (we have renumbered the appendices as this reference now comes earlier).  We would be happy to move the inclusion criteria to create a table in the main text if you feel this is appropriate, but it would make the paper longer. 

Why include magnesium and iron?

Authors’ reply: WHO had asked us to carry out all three scoping reviews at the same time.  As there was considerable overlap between the titles and abstracts located by the 3 searches (those for vitamin A, iron and magnesium) it was more efficient to run combined searches and assess inclusion of the full set of studies at the same time.  We have now explained this more clearly in the text (lines 200-203).

Describe how you assessed the risk of bias for each type of study. Include topic.

Authors’ reply: we have added a section called Risk of bias assessment to clarify (lines 265 to 271)

RESULTS Need to include more tables to understand the study

Authors’ reply: Good idea, we have added a tabulation of the studies found relevant to the health-based questions of the review (now Table 1, page 10). 

DISCUSSION Same results. It ends without the application of this study... The study has no conclusion.

Authors’ reply: thank you, we have added a set of Conclusions (page 15, lines 521 onwards).  Additionally, we see the conclusion as the excel spreadsheet – the dataset that was produced through the research (and is included as a supplementary file). 

REFERENCES Adapt to the magazine's format

Authors’ reply: completed, the format should now be correct

Yours sincerely, Lee Hooper (on behalf of all the authors)

  1. Imdad A, Mayo-Wilson E, Herzer K, Bhutta ZA. Vitamin A supplementation for preventing morbidity and mortality in children from six months to five years of age. The Cochrane database of systematic reviews. 2017;3:CD008524. https://dx.doi.org/10.1002/14651858.CD008524.pub3